# Positive Correlation of Peripheral CD8^+^ T Lymphocytes with Immune-Related Adverse Events and Combinational Prognostic Value in Advanced Non-Small Cell Lung Cancer Patients Receiving Immune Checkpoint Inhibitors

**DOI:** 10.3390/cancers14153568

**Published:** 2022-07-22

**Authors:** Kan Wu, Bing Xia, Jing Zhang, Xin Li, Shaoyu Yang, Minna Zhang, Lucheng Zhu, Bing Wang, Xiao Xu, Shenglin Ma, Xueqin Chen

**Affiliations:** 1Department of Thoracic Oncology, Affiliated Hangzhou Cancer Hospital, Zhejiang University School of Medicine, Hangzhou 310002, China; wukanwukan@126.com (K.W.); xb0918@hotmail.com (B.X.); zmn8073@163.com (M.Z.); zhulucheng1@outlook.com (L.Z.); wangbingdrice@foxmail.com (B.W.); xuxiao116@sina.com (X.X.); 2Department of Thoracic Oncology, The Forth School of Clinical Medicine, Zhejiang Chinese Medical University, Hangzhou 310053, China; zhangjing1224ug@163.com; 3Key Laboratory of Clinical Cancer Pharmacology and Toxicology Research of Zhejiang Province, Department of Thoracic Oncology, Affiliated Hangzhou First People’s Hospital, Zhejiang University School of Medicine, Hangzhou 310006, China; lixtea@gmail.com (X.L.); yangshaoy@126.com (S.Y.); 4Cancer Center, Zhejiang University, Hangzhou 310058, China

**Keywords:** immune-related adverse events, baseline peripheral CD8^+^ T lymphocytes, non-small cell lung cancer, prognostic

## Abstract

**Simple Summary:**

Immune checkpoint inhibitors are widely used in clinical practice and have demonstrated remarkable efficacy in advanced non-small cell lung cancer (NSCLC). However, their use is also commonly accompanied by immune-related adverse events (irAEs), and markers that are able to predict the onset of irAEs represent an urgent need. In this study, we found that the baseline level of peripheral CD8^+^ T lymphocytes was the independent risk factor of the onset of irAEs, and it was associated with longer survival in advanced NSCLC patients treated with PD-1/PD-L1 inhibitors. Furthermore, the study showed the combinational predictive value of baseline CD8^+^ T lymphocytes and the onset of irAEs for the clinical outcomes.

**Abstract:**

Immune checkpoint inhibitors (ICIs) therapy has revolutionized the treatment patterns of non-small cell lung cancer (NSCLC). However, patients treated with ICIs may experience immune-related adverse events (irAEs). Markers that could predict the onset of irAEs are still unclear. Here, we report the possible correlation of baseline peripheral lymphocytes with irAEs and clinical outcomes in advanced NSCLC patients receiving ICIs. A total of 109 advanced NSCLC patients treated with ICIs from April 2017 to January 2021 were analyzed retrospectively. Logistic and Cox regression analyses was applied to evaluate independent risk factors for irAEs, progression-free survival (PFS), and overall survival (OS). Among these patients, 55 (50.5%) patients experienced irAEs. The level of CD8^+^ T lymphocytes at baseline was the independent risk factor for the onset of irAEs (*p* = 0.008). A higher level of CD8^+^ T lymphocytes was associated with longer PFS (11.0 months vs. 3.0 months, *p* < 0.001) and OS (27.9 months vs. 11.7 months, *p* = 0.014). Furthermore, patients who had higher baseline CD8^+^ T lymphocytes and experienced irAEs had a longer PFS (18.4 months vs. 2.2 months, *p* < 0.001) and OS (32.8 months vs. 9.0 months, *p* = 0.001) than those who had lower CD8+ T lymphocytes and no irAEs. Our study highlights the value of baseline peripheral CD8^+^ T lymphocytes as a predictive factor for irAEs in advanced NSCLC patients receiving ICIs. In addition, patients who have higher baseline CD8^+^ T lymphocytes and experience irAEs would have a superior PFS and OS.

## 1. Introduction

Lung cancer is a commonly diagnosed cancer and the leading cause of cancer-related death worldwide [1]. Non-small cell lung cancer (NSCLC) accounts for approximately 85% of all cases, and more than half of these are diagnosed after the cancer has already metastasized [2]. In recent years, immune checkpoint inhibitors (ICIs), including antibodies to programmed cell death-1 (PD-1) and programmed cell death ligand 1 (PD-L1), have revolutionized the treatment of several types of malignancy, including head and neck carcinoma [3], hepatocellular carcinoma [4], esophageal carcinoma [5], colorectal carcinoma, [6] and NSCLC [7,8], and show superior median overall survival (OS) and long-term survival compared to standard chemotherapy for some special groups [9,10]. Although ICIs therapy is better tolerated in NSCLC patients than traditional chemotherapy, patients treated with ICIs may still experience a wide range of side effects, which are called immune-related adverse events (irAEs). IrAEs occur in any organ, most frequently involving skin, endocrine organs, the lung, and the liver [11]. The incidence of all grade irAEs (according to Common Terminology Criteria for Adverse Events (CTCAE)) has been reported from 58% to 69%. Most of the irAEs are mild or moderate (grade 1 or grade 2), and the incidence of grade 3–5 irAEs is about 7–13% in NSCLC [12,13,14,15]. Although severe irAEs remain rare, they could become life-threatening if not recognized early and accurately and managed appropriately [16]. Therefore, effective predictive markers for identify risk factors of irAEs represent an urgent need in clinic.

The mechanisms for irAEs have still not been elucidated. Some potential mechanisms include increasing T-cell activity against antigens that are present in tumors and healthy tissue, increasing levels of preexisting autoantibodies, and an increase in the level of inflammatory cytokines [17]. The enhancement of systemic T-cell activity by ICIs causes a loss of immune tolerance in various organs, resulting in irAEs [17,18]. There is a complex interplay between the immune system and tumor. It has been reported that T-cell tumor infiltrating lymphocytes (TILs) and circulating T-cell subpopulations are associated with clinical outcomes in many cancer types, including NSCLC [19,20,21,22,23]. Up to now, markers that could predict the onset of irAEs are still unclear. It is suspected that several factors including tumor mutational burden, TILs, PD-L1, gut bacterial diversity, and cytokines may be linked to the occurrence of irAEs [24]. Baseline peripheral lymphocytes have shown a great potential as a predictor of irAEs because of the advantages in practically noninvasive and dynamic monitoring. However, studies in this area are still limited.

The present study explored the potential predictive value of clinical characteristics, especially baseline subsets of lymphocytes on the onset of irAEs, and investigated the factors associated with clinical outcomes in advanced NSCLC patients receiving ICIs-based therapy.

## 2. Materials and Methods

### 2.1. Patients and Data Collection

Patients histologically or cytologically confirmed advanced NSCLC (IIIB/IV), treated with ICIs at a single institution from April 2017 to January 2021, were analyzed retrospectively. Other inclusion criteria were as follow: (1) Eastern Cooperative Oncology Group (ECOG) performance status (PS) 0–2; (2) complete medical records, including baseline level of lymphocytes subsets; (3) treatment with anti-PD1/anti-PD-L1 drugs or anti-PD1/anti-PD-L1 drugs in combination with chemotherapy or antiangiogenesis therapy; and (4) all patients received at least 3 months of immunotherapy unless disease progression or unacceptable toxicity occurred. The PD1/PD-L1 inhibitors were administered intravenously, according to the instructions.

Information including patient demographics, clinical characteristics, treatment patterns, and baseline subsets of lymphocytes were collected. The baseline levels of lymphocytes subsets (defined as the most recent records within 1 week before immunotherapy initiation), including CD4^+^ T lymphocytes, CD8^+^ T lymphocytes, and regulatory T lymphocytes, were recorded. A total of 109 advanced NSCLC patients treated with ICIs were included in the study. Overall, six ICIs were assessed, namely pembrolizumab (n = 46), nivolumab (n = 28), durvalumab (n = 5), sintilimab (n = 11), toripalimab (n = 14), and atelizumab (n = 5). At the time of analysis, the median follow-up time was 9.0 months (0.1–47.4), and the median treatment duration was 3.6 months (0.1–39.5). The clinical characteristics of the enrolled patients are summarized in Table 1. The median age was 65 years (36–85), 19.3% of participants were female, 42.2% of patients were never-smokers, and 84.4% of patients had an ECOG PS of 0–1. Most patients had adenocarcinoma (55.0%), and most had no or undetected common driver gene (including epidermal growth factor receptor (EGFR), anaplastic lymphoma kinase (ALK), and V-ros UR2 sarcoma virus oncogene homolog 1 (ROS1)) alteration (91.7%). The disease stage was IIIB in 6.4% of the patients and IV in 93.6% of the patients. In this cohort, 73 (67.0%) patients were tested for PD-L1 expression in tumor tissue, and 76.7% of those were PD-L1 positive. A total of 41 (37.6%) patients accepted ICIs as first-line treatment. About 42.2% of patients received ICIs monotherapy, while 57.8% of patients received ICIs combined with chemotherapy or antiangiogenesis therapy. The median baseline level of the CD4^+^ T lymphocytes count was 266 (range 37–1008) M/L, the median baseline level of the CD8^+^ T lymphocytes count was 288 (range 25–1195) M/L, and the median baseline level of regulatory T lymphocytes count was 17 (range 3–63) M/L.

### 2.2. Study Assessment

Tumor response evaluation was performed every 6–9 weeks, according to the response evaluation criteria in solid tumor (RECIST) (version 1.1; https://recist.eortc.org/recist-1-1-2/, accessed on 26 April 2017). Baseline measurements were defined as those taken within 4 weeks before receiving ICIs. Progression-free survival (PFS) was defined as the time from the date of initial administration of ICIs to first disease progression or death due to any cause, censored at the date of the latest survival information available or initiation of other treatment. OS was defined as time from the first day of ICIs to death by any cause, or censored at the date of the latest survival information available.

Toxicity was assessed by CTCAE (version 5.0; https://ctep.cancer.gov/protocoldevelopment/electronic_applications/docs/ctcae_v5_quick_reference_5x7.pdf, accessed on 6 March 2018). IrAEs were defined as AEs with a potential immunologic basis that required more frequent monitoring and potential intervention with corticosteroids and other immunomodulatory agents, such as pneumonitis, hepatic dysfunction, thyroid disorder, rash, and other conditions. Improvement or resolution of irAEs was also assessed by the investigators.

### 2.3. Statistical Analysis

In the statistical analysis, descriptive statistics were calculated for categorical variables and continuous variables. A univariate logistic regression model (Enter) was applied to explore the correlation of the demographic, clinical data, baseline level of lymphocytes subsets, and the onset of irAEs. We examined the Pearson’s correlation between the median baseline level of CD8^+^ T lymphocytes and the incidence of irAEs. Univariate Cox proportional hazard regression model (Enter) was performed to clarify the risk factors for PFS and OS. Multivariate logistic and Cox analysis were performed by using the logistic/Cox regression model (Enter) containing all of the variables that attained or trended toward univariate statistical significance (*p* < 0.10). PFS and OS curves were calculated by using the Kaplan–Meier method, and the log-rank test was employed to assess differences. The results were considered statistically significant when the two-sided *p*-value was < 0.05. All data analyses were performed with SPSS version 21.0 software (SPSS Inc., Chicago, IL, USA) and GraphPad Prism 8.0 (GraphPad Inc., La Jolla, CA, USA).

## 3. Results

### 3.1. Profiles of IrAEs

A total of 55 patients (50.5%) experienced at least one type of irAE, and 38 patients (34.9%) developed multiple irAEs. Ninety-three patients (85.3%) experienced mild and/or moderate (grade 1 and/or grade 2) irAEs, while grade 3–5 irAEs occurred in 16 patients (14.7%) in the study. Treatment related death was observed in 2 patients (1.8%). A total of 148 irAEs were observed in all patients, and 104 irAEs (70.3%) were resolved by the end of the observation. The most common any-grade irAEs were rash (16.5%), hypothyroidism (15.6%), pruritus (13.8%), pneumonitis (12.8%), hyperthyroidism (11.9%), alanine transaminase increase (11.9%), gamma-glutamyltransferase increase (11.9%), aspartate aminotransferase increase (9.2%), and pyrexia (6.4%). The most common grade 3–5 irAEs were pneumonitis in four patients (3.7%) and gamma-glutamyltransferase increase in three patients (2.8%) (Figure 1 and Table 2). The median time to onset was 6.9 weeks and median time to resolution was 3.4 weeks for all irAEs (Appendix A). The majority of new irAEs occurred from cycles 1 to 3 in all patients, and few new events occurred after 15 months (Appendix A).

### 3.2. Association of CD8^+^ T Lymphocytes at Baseline with the Occurrence of IrAEs

Demographic, clinical, treatment characteristics and baseline levels of lymphocytes subsets were analyzed under univariate and multivariate logistics analyses as candidate factors predicting irAEs. Patients were divided into high and low groups according to the median value of CD4^+^ T lymphocytes (266 M/L), CD8^+^ T lymphocytes (288 M/L), and regulatory T lymphocytes (17 M/L). According to the univariate logistics analysis, the high-CD8^+^-T-lymphocytes group had a significantly higher incidence of irAEs compared with the low-CD8^+^-T-lymphocytes group (OR = 2.975, 95% CI = 1.365–6.484, and *p* = 0.006). On the contrary, patients with an ECOG PS of 2 tended to have a lower incidence of irAEs than those with an ECOG PS of 0 or 1 (OR = 0.350, 95% CI = 0.114–1.074, and *p* = 0.066). Similarly, patients who harbored common driver gene alterations tended to have a lower incidence of irAEs (OR = 0.253, 95% CI = 0.050–1.280, and *p* = 0.097). No significant correlations were observed between irAEs and other lymphocytes’ subsets and clinical characteristics (Table 3).

The multivariate logistic regression analysis showed that the level of CD8^+^ T lymphocytes was the independent predictor for the occurrence of irAEs (OR = 2.953, 95% CI = 1.324–6.587, and *p* = 0.008; Table 3). The incidence of irAEs was more frequent in the high CD8^+^ T lymphocytes group (63.6%) than in the group with low CD8^+^ T lymphocytes (37.0%, *p* < 0.001). Furthermore, we divided patients into six groups according to the baseline level of CD8+ T lymphocytes (20 cases per group in Groups 1 to 5, and 9 cases in Group 6). Our analysis revealed a significant positive correlation between the incidence of irAEs and the median baseline level of CD8^+^ T lymphocytes (Pearson correlation coefficient, R = 0.817; *p* = 0.047; Appendix A).

### 3.3. Factors Predictive of Clinical Outcomes

Among the overall population, the median PFS and median OS were 5.5 months (95% CI = 3.2–7.8 months) and 16.8 months (95% CI = 9.5–24.1 months), respectively. The univariable and multivariable analysis results are showed in Table 4A,B. In the univariate analysis, the group with high CD8^+^ T lymphocytes had a significantly longer PFS and OS than that of the group with the low CD8^+^ T lymphocytes (HR = 0.386, 95% CI = 0.237–0.629, and *p* < 0.001 vs. HR = 0.485, 95% CI = 0.273–0.862, and *p* = 0.014, respectively). Patients with an ECOG PS of 2 had a shorter PFS and OS than those with an ECOG PS of 0 or 1 (HR = 2.146, 95% CI = 1.208–3.812, and *p* = 0.009 vs. HR = 1.948, 95% CI = 1.004–3.781, and *p* = 0.049). Patients who received ICIs as later-line therapy had a shorter PFS and OS than those receiving ICIs as first-line therapy (HR = 3.211, 95% CI = 1.857–5.551, and *p* < 0.001 vs. HR = 2.413, 95% CI = 1.234–4.715, and *p* = 0.010). Patients with irAEs had a significantly longer PFS and OS compared with those without irAEs (HR = 0.368, 95% CI = 0.227–0.596, and *p* < 0.001 vs. HR = 0.439, 95% CI = 0.247–0.782, and *p* = 0.005). In addition, the administration of combination therapy was associated with a longer PFS (HR = 0.636, 95% CI = 0.385–1.053, and *p* = 0.078), but there was no significant difference in OS. Patients who harbored common driver gene alterations had a shorter OS (HR = 2.474, 95% CI = 1.042–5.872, and *p* = 0.040).

The multivariate analysis was performed to identify independent prognostic factors related to PFS and OS (Table 4A,B). Variables with a *p*-value ≤ 0.10 in the univariate analysis were included in the multivariable analyses. High-CD8^+^-T-lymphocytes group still had a longer PFS than the low-CD8^+^-T-lymphocytes group (HR = 0.364, 95% CI = 0.217–0.612, and *p* < 0.001). Patients with a poor ECOG PS had a shorter PFS than those with a good ECOG PS (HR = 1.882, 95% CI = 1.036–3.420, and *p* = 0.038). Receiving ICIs as later-line therapy was associated with a shorter PFS (HR = 3.479, 95% CI = 1.906–6.349, and *p* < 0.001) than receiving ICIs as the first-line ICIs treatment. Patients with irAEs had a significantly longer PFS compared with those without irAEs (HR = 0.344, 95% CI = 0.204–0.578, and *p* < 0.001). However, there were no prognostic factors of OS according to the multivariate Cox analysis.

### 3.4. Association among Baseline CD8^+^ T Lymphocytes, IrAEs and Clinical Outcomes

The Kaplan–Meier curve analysis demonstrated a significantly higher median PFS and OS in the group with high CD8^+^ T lymphocytes than in the group with low CD8^+^ T lymphocytes (median, 11.0 months vs. 3.0 months, *p* < 0.001; 27.9 months vs. 11.7 months, *p* = 0.014) (Figure 2A,B). Patients with irAEs had a significantly longer PFS and OS compared with the no-irAEs group (median, 12.5 months vs. 3.2 months, *p* < 0.001; 27.9 months vs. 10.5 months, *p* = 0.005) (Figure 2C,D). We explored the OS and PFS by high CD8^+^ T lymphocytes and/or the occurrence of irAEs (Figure 2E,F). All patients were divided into four groups: patients who developed irAE with high or low CD8^+^ T lymphocytes were assigned to Group A or B, while patients not presenting irAE with high or low CD8^+^ T lymphocytes were assigned to Group C or D. The PFS of Group A was significantly longer than that of Groups B, C, and D (*p* = 0.032, *p* = 0.040, and *p* < 0.001). However, the difference in OS was observed only between Groups A and D (*p* = 0.001), and there was no difference in the OS between patients with one favorable factor (high CD8^+^ T lymphocytes or irAEs) and those have both favorable factors.

## 4. Discussion

Historically, survival rates for advanced NSCLC were disappointingly low, and treatment-related toxicities were extremely significant. Although ICIs have achieved important progress in driver-gene negative advanced NSCLC patients with acceptable adverse events in recent years, grade 3–5 irAEs could still occur [15,25]. Therefore, effective predictive factors are urgently needed for the identification of advanced NSCLC patients who may suffer from irAEs. Herein, our study demonstrated that baseline peripheral CD8^+^ T lymphocytes were significantly associated with occurrence of irAEs. Meanwhile, the median PFS and OS of the patients with higher baseline CD8^+^ T lymphocytes was significantly better than that of patients with lower baseline CD8^+^ T lymphocytes. Furthermore, the study also showed that the combination of baseline CD8^+^ T lymphocytes and the onset of irAEs could be a more valuable prognostic factor for the clinical outcomes. 

In our study, approximately 50% of patients presented any-grade irAEs, and about 15% developed ≥ grade 3 irAEs. Skin-, endocrine-, hepatic-, and pulmonary-related toxicities were the most commonly reported irAEs, while pneumonitis was the most common grade 3–5 irAE. These findings are similar to those of the existing literature on the lung cancer patients treated with ICIs [12,13,14,15]. A variety of markers from the tumor microenvironment, circulating blood, target organs, or clinical factors have been reported to be associated with the onset of irAEs. However, none of them has been identified as a definitive predictive marker. Previous studies indicated that cytokines were most likely responsible for irAEs, as the inhibition of cytokines could rapidly reverse that condition [26]. A recent study showed that baseline and dynamic IL-10 levels were significantly and independently related to a higher incidence of irAEs (OR = 5.318, 95% CI = 1.174–24.081, and *p* = 0.030) [27]. In addition, the literature showed that autoantibodies, including thyroglobulin antibody [28], and gut microbiota such as Faecalibacterium [29] were also associated with the occurrence of irAEs. Moreover, the neutrophil lymphocyte ratio (NLR) (OR = 1.046; CI 95% = 1.006–1.088), a higher level of PD-L1 expression (OR = 2.009; 95% CI = 1.03–3.921), and smoking status (OR = 1.249; 95% CI = 1.021–1.528) were significantly associated with irAEs in a meta-analyses [30]. However, in our study, no statistical difference was observed between smoking status, expression of PD-L1, and irAEs. Possible reasons are the partial loss of PD-L1 data and selection bias of patients for receiving ICIs therapy in the real world. 

Interestingly, we observed that baseline peripheral CD8^+^ T lymphocytes were associated with irAEs. Although the mechanisms underlying the onset of irAEs remain unclear, some studies have revealed that the potential mechanism is related to T cells. One study showed that, in patients treated with ICIs, the expansion of specific CD8^+^ T cell clones was associated with CTCAE grade 2 and 3 irAEs [31]. Furthermore, a recent study research reported a significant accumulation of CD8^+^ T cells with markers of cytotoxicity and proliferation in tissue biopsy samples from patients with colitis, suggesting that a significant subset of T-cell clones associated with irAEs were pre-existent [32]. Additional studies reported T-cell TILs were associated with hepatitis and pneumonitis [33,34]. It may be hypothesized that activated T-cell TILs are a hallmark of irAEs, and activated T cells kill tumor cells and may attack normal human tissue cells, forming ICIs-related toxicities. Compared with TILs, peripheral lymphocytes are more convenient and noninvasive. To our knowledge, this is the first study to investigate the predictive value of peripheral CD8^+^ T cells for irAEs. Of course, these preliminary results warrant further research on a larger patient cohort.

It is now widely recognized that there is the correlation between T lymphocytes and clinical outcomes in cancer patients treated with immunotherapy. IMpassion130, a phase III multicenter, randomized, double-blind, placebo-controlled trial in metastatic triple-negative breast cancer patients showed that high CD8^+^ TILs expression was significantly associated with improvements in OS and PFS [19]. A meta-analysis encompassing 14,395 NSCLC patients revealed that high CD8^+^ TILs were associated with improved prognosis predictions for OS (3-year OS AUC: 0.659; 5-year OS AUC: 0.665) [21]. In a study of patients with melanoma (n = 43) and non-squamous NSCLC (n = 40), patients with high circulating central memory/effector T-cell ratios experienced extended PFS compared with patients with low ratios [22]. In a study of 29 NSCLC patients treated with ICIs, the early proliferation of PD-1^+^ CD8^+^ T cells following ICIs treatment may be associated with clinical response [23]. Similarly, in our study, we observed that high baseline peripheral CD8^+^ T lymphocytes were associated with improved PFS and OS. Although baseline CD8^+^ T lymphocytes were not an independent prognostic factor of OS, these results also indicate that ICIs therapy should be used with caution in patients with low baseline peripheral CD8^+^ T lymphocytes. Certainly, the molecular feature of CD8^+^ T lymphocytes’ subsets may have preferable predictive value, such as its subsets proportion and PD-L1 expression status, which would be of further concern in the future. Furthermore, existing evidence has showed the prognosis value of irAEs in NSCLC patients treated with ICIs. A retrospective study of 134 patients with advanced or recurrent NSCLC who were treated with nivolumab revealed that the development of irAEs was positively associated with PFS (*p* = 0.03) and OS (*p* = 0.003) [35]. A prospective observational study of 38 NSCLC patients reported a correlation between irAEs and efficacy in NSCLC patients treated with nivolumab [36]. Similarly, the current study has provided evidence of the correlation between irAEs and survival in NSCLC treated with ICIs. In addition, our data revealed detailed information about the onset and resolution time of irAEs, which may provide important clues about the mechanisms linking irAEs with the efficacy of ICIs. The relationship of irAEs’ site, grade, timing of onset, and clinical outcomes deserves a further analysis in the future. Moreover, the current study has also explored the combined predictive effect of irAEs and the peripheral blood CD8^+^ T lymphocytes and found that patients with both high CD8^+^ T lymphocytes and irAEs have the best PFS. Thus, an evaluation of baseline CD8^+^ T lymphocytes and irAEs may provide meaningful information for efficacy prediction in immunotherapy.

We acknowledge a number of potential limitations in this study. Firstly, the cohort population was heterogenous in the disease context (first-line treatment vs. later-line treatment), and this might affect the outcomes [37]. Furthermore, the retrospective nature and the relatively small sample size limit the generalizability of the study. Although the results of our study are interesting, the combination predictive value of the CD8^+^ T lymphocytes and irAEs requires further validation by prospective large sample studies. Fortunately, there was no missing survival and lymphocytes test data in our analysis. Despite these limitations, the study is unique as the first to explore the predictive value of peripheral CD8^+^ T cells for irAEs and the combinational predictive value for the clinical outcomes in advanced NSCLC treated with ICIs.

## 5. Conclusions

Our data indicated that the baseline peripheral CD8^+^ T lymphocytes were correlated with the onset of irAEs and clinical outcomes. Importantly, the study revealed the combinational prognostic value of peripheral CD8^+^ T lymphocytes and irAEs in advanced NSCLC patients treated with PD-1/PD-L1 inhibitors. An evaluation of baseline CD8^+^ T lymphocytes and irAEs may be a convenient method to identify clinical outcomes in a timely manner. Certainly, these preliminary results warrant further research.

## Figures and Tables

**Figure 1 cancers-14-03568-f001:**
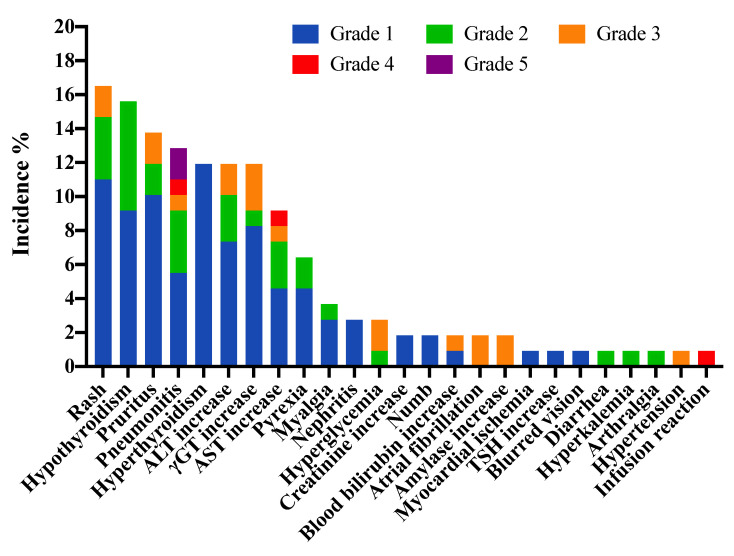
Profiles of immune-related adverse events among 109 advanced non-small cell lung cancer patients. Abbreviation: ALT, alanine transaminase; AST, aspartate aminotransferase; γGT, gamma-glutamyltransferase.

**Figure 2 cancers-14-03568-f002:**
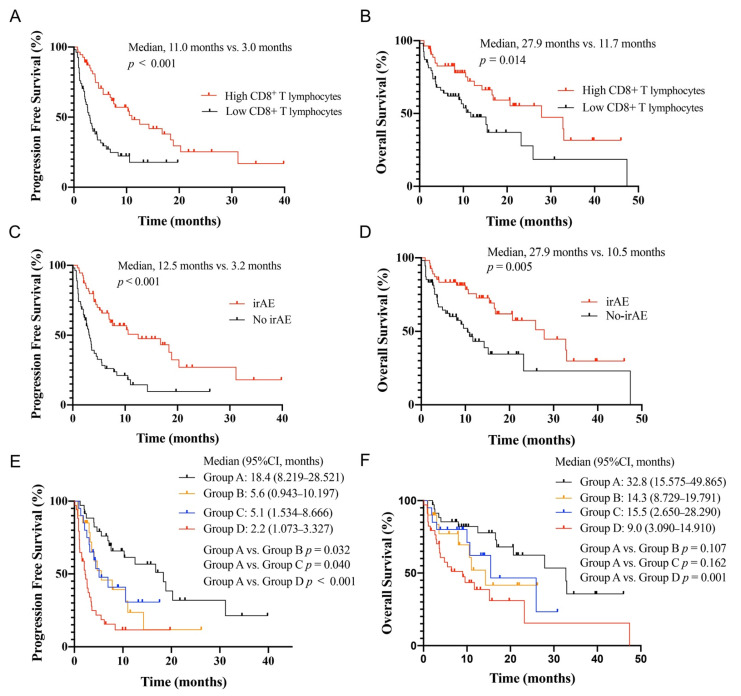
Association between baseline peripheral blood CD8^+^ T lymphocytes, immune-related adverse events (irAEs), and survival in non-small cell lung cancer patients treated with ICIs. PFS (**A**) and OS (**B**) curves of patients according to baseline peripheral blood CD8^+^ T lymphocytes. PFS (**C**) and OS (**D**) curves of patients stratified according to the onset of irAEs. PFS (**E**) and OS (**F**) curves of patients stratified according to baseline peripheral blood CD8^+^ T lymphocytes and irAEs. Group A, high CD8^+^ T lymphocytes and irAEs; Group B, low CD8^+^ T lymphocytes and irAEs; Group C, high CD8^+^ T lymphocytes and no irAEs; and Group D, low CD8^+^ T lymphocytes and no irAEs. Abbreviation: ICIs, immune checkpoint inhibitors; PFS, progression-free survival; OS, overall survival.

**Table 1 cancers-14-03568-t001:** Baseline characteristics of 109 advanced non-small cell lung cancer patients.

Characteristics	Subsets	No.	%
Gender	Male	88	80.7%
	Female	21	19.3%
Age (years)	Median	65	
	Range	36–85	
ECOG PS score	0–1	92	84.4%
	2	17	15.6%
Smoking status	Current/former	63	57.8%
	Never	46	42.2%
Histology	Adenocarcinoma	60	55.0%
	Squamous carcinoma	46	42.2%
	Large cell carcinoma	3	2.8%
Stage	Ⅳ	102	93.6%
	ⅢB	7	6.4%
PD-L1 states test	≥50%	23	21.1%
	1–49%	33	30.3%
	<1%	17	15.6%
	Unknown	36	33.0%
Driver gene alteration	EGFR/ALK/ROS1 positive	9	8.3%
	Negative/unknown	100	91.7%
Treatment line	First	41	37.6%
	Second	39	35.8%
	Third	15	13.8%
	Fourth or later line	14	12.8%
Combination treatment	Yes	63	57.8%
	No	46	42.2%

ECOG PS, Eastern Cooperative Oncology Group performance status; PD-L1, programmed cell death ligand 1; EGFR, epidermal growth factor receptor; ALK, anaplastic lymphoma kinase; ROS1, V-ros UR2 sarcoma virus oncogene homolog 1.

**Table 2 cancers-14-03568-t002:** Immune-related adverse events (irAEs) by characteristics and resolution rates.

IrAEs Category	No. of Patients, n = 109		No. of Resolved Events, n = 148, (%)
	Any Grade, n (%)	Grade 3–5, n (%)	
Any irAEs	55 (50.5%)	16 (14.7%)	104 (70.3%)
Single site irAEs	17 (15.6%)	/	
Multiple site irAEs	38 (34.9%)	/	
Pulmonary			
Pneumonitis	14 (12.8%)	4 (3.7%)	8 (57.1%)
Cardiovascular			
Myocardial ischemia	1 (0.9%)	0	1 (100.0%)
Atrial fibrillation	2 (1.8%)	2 (1.8%)	1 (50.0%)
Hypertension	1 (0.9%)	1 (0.9%)	0
Gastrointestinal			
Diarrhea	1 (0.9%)	0	1 (100.0%)
Amylase increase	2 (1.8%)	2 (1.8%)	1 (50.0%)
Hepatic			
ALT increase	13 (11.9%)	2 (1.8%)	11 (84.6%)
AST increase	10 (9.2%)	2 (1.8%)	7 (70.0%)
γGT increase	13 (11.9%)	3 (2.8%)	10 (76.9%)
Blood bilirubin increase	2 (1.8%)	1 (0.9%)	2 (100.0%)
Renal			
Nephritis	3 (2.8%)	0	1 (33.3%)
Creatinine increase	2 (1.8%)	0	1 (50.0%)
Hyperkalemia	1 (0.9%)	0	1 (100.0%)
Musculoskeletal			
Myalgia	4 (3.7%)	0	3 (75.0%)
Arthralgia	1 (0.9%)	0	1 (100.0%)
Endocrine			
Hypothyroidism	17 (15.6%)	0	4 (23.5%)
TSH increase	1 (0.9%)	0	1 (100.0%)
Hyperthyroidism	13 (11.9%)	0	12 (92.3%)
Hyperglycemia	3 (2.8%)	2 (1.8%)	1 (33.3%)
Skin			
Rash	18 (16.5%)	2 (1.8%)	15 (83.3%)
Pruritus	15 (13.8%)	2 (1.8%)	13 (86.7%)
Eye			
Blurred vision	1 (0.9%)	0	1 (100.0%)
Neurology			
Numb	2 (1.8%)	0	0
Others			
Infusion reaction	1 (0.9%)	1 (0.9%)	1 (100.0%)
Pyrexia	7 (6.4%)	0	7 (100.0%)

ALT, Alanine transaminase; AST, Aspartate aminotransferase; γGT, Gamma-Glutamyltransferase; TSH, Thyroid Stimulating Hormone.

**Table 3 cancers-14-03568-t003:** Univariate and multivariate analyses for the risk factors of immune-related adverse events (irAEs).

Variable	Category	Univariate			Multivariate		
		OR	95% CI	*p*-Value	OR	95% CI	*p*-Value
Gender	Male	1.151	0.444–2.985	0.772			
Age (years)	≥65 years	1.292	0.606–2.752	0.507			
Histology	Non-adenocarcinoma	1.630	0.762–3.487	0.208			
ECOG PS score	2	0.350	0.114–1.074	0.066	0.355	0.109–1.163	0.087
Smoking status	Former/current	0.656	0.305–1.410	0.280			
Treatment line	≥Second	0.593	0.271–1.299	0.192			
Driver gene alterlation	EGFR/ALK/ROS1 positive	0.253	0.050–1.280	0.097	0.308	0.055–1.709	0.178
Combination treatment	Yes	0.833	0.389–1.784	0.639			
PD-L1 states test	≥50%	1.362	0.539–3.440	0.513			
CD4^+^ T lymphocytes	≥266 M/L	1.392	0.655–2.957	0.390			
CD8^+^ T lymphocytes	≥288 M/L	2.975	1.365–6.484	0.006	2.953	1.324–6.587	0.008
Regulatory T lymphocytes	≥17 M/L	0.773	0.364–1.640	0.502			

ECOG PS, Eastern Cooperative Oncology Group performance status; PD-L1, programmed cell death ligand 1; EGFR, epidermal growth factor receptor; ALK, anaplastic lymphoma kinase; ROS1, V-ros UR2 sarcoma virus oncogene homolog 1.

**Table 4 cancers-14-03568-t004:** (A) Cox analysis of influencing factors for progression-free survival of 109 lung cancer patients treated with immunotherapy. (B) Cox analysis of influencing factors for overall survival of 109 lung cancer patients treated with immunotherapy.

**(A)**							
**Variable**	**Category**	**Univariate**			**Multivariate**		
		**HR**	**95% CI**	***p*-Value**	**HR**	**95% CI**	***p*-Value**
Gender	Male	1.091	0.584–2.040	0.785			
Age (years)	≥65 years	0.975	0.612–1.551	0.913			
Histology	Non-adenocarcinoma	1.114	0.700–1.774	0.648			
ECOG PS score	2	2.146	1.208–3.812	0.009	1.882	1.036–3.420	0.038
Smoking status	Former/current	1.284	0.793–2.080	0.309			
Treatment line	≥Second	3.211	1.857–5.551	<0.001	3.479	1.906–6.349	<0.001
Driver gene alterlation	EGFR/ALK/ROS1 positive	1.934	0.831–4.500	0.126			
Combination treatment	Yes	0.636	0.385–1.053	0.078	0.638	0.369–1.102	0.107
PD-L1 states test	≥50%	0.877	0.714–1.078	0.213			
CD4^+^ T lymphocytes	≥266 M/L	0.850	0.533–1.356	0.496			
CD8^+^ T lymphocytes	≥288 M/L	0.386	0.237–0.629	<0.001	0.364	0.217–0.612	<0.001
Regulatory T lymphocytes	≥17 M/L	0.911	0.572–1.452	0.696			
IrAEs	Yes	0.368	0.227–0.596	<0.001	0.344	0.204–0.578	<0.001
**(B)**							
**Variable**	**Category**	**Univariate**			**Multivariate**		
		**HR**	**95% CI**	***p*-Value**	**HR**	**95% CI**	***p*-Value**
Gender	Male	1.379	0.615–3.094	0.436			
Age (years)	≥65 years	0.958	0.548–1.673	0.879			
Histology	Non-adenocarcinoma	1.095	0.627–1.910	0.750			
ECOG PS score	2	1.948	1.004–3.781	0.049	1.596	0.796–3.200	0.188
Smoking status	Former/current	1.248	0.692–2.252	0.462			
Treatment line	≥Second	2.413	1.234–4.715	0.010	1.952	0.977–3.902	0.058
Driver gene alterlation	EGFR/ALK/ROS1 positive	2.474	1.042–5.872	0.040	1.890	0.775–4.609	0.162
Combination treatment	Yes	0.693	0.367–1.310	0.259			
PD-L1 states test	≥50%	0.883	0.702–1.110	0.287			
CD4^+^ T lymphocytes	≥266 M/L	0.674	0.385–1.178	0.166			
CD8^+^ T lymphocytes	≥288 M/L	0.485	0.273–0.862	0.014	0.647	0.348–1.202	0.169
Regulatory T lymphocytes	≥17 M/L	0.655	0.373–1.149	0.140			
IrAEs	Yes	0.439	0.247–0.782	0.005	0.599	0.320–1.120	0.109

ECOG PS, Eastern Cooperative Oncology Group performance status; PD-L1, programmed cell death ligand 1; EGFR, epidermal growth factor receptor; ALK, anaplastic lymphoma kinase; ROS1, V-ros UR2 sarcoma virus oncogene homolog 1; IrAEs, immune-related adverse events.

## Data Availability

Data are not publicly available due to privacy and ethical restrictions.

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
