# Peer review of "Positive Correlation of Peripheral CD8+ T Lymphocytes with Immune-Related Adverse Events and Combinational Prognostic Value in Advanced Non-Small Cell Lung Cancer Patients Receiving Immune Checkpoint Inhibitors"

_cancers, 2022, doi:10.3390/cancers14153568_

Round 1

Reviewer 1 Report

Manuscript Revision (cancers-1800258)

The manuscript entitled ”Positive Correlation of Peripheral CD8+ T Lymphocytes with Immune-Related Adverse Events and Combinational Prognostic Value in Advanced Non-Small Cell Lung Cancer Patients Receiving Immune Checkpoint Inhibitors”

Is well written and gives an interesting contribution to immunotherapy outcome knowledge, offering a good cue for future studies.

Nevertheless, I would suggest some minor modifications to improve the presentation of the results and give more weight to the manuscript.

Abstract.

Lines 41-43: as it is written it seems contradictory, I would suggest highlighting that it is the combination of high CD8+ T and immune-related adverse events to represent a prognostic tool for the assessment of clinical outcomes after immunotherapy.

Lines 45-47: please write differently, because it is a little confusing for readers.

Introduction.

Line 66: what about grade 1 and 2 adverse events? Please write just a few words to mention.

Lines 83-86: please review the English language.

Lines 83-86: is this coming from your observation or does it come from literature? In the latter, please cite the Reference.

Materials and Methods.

Please review the English language.

Paragraph 2.1.

Line 99: which other therapies are you referring to? Please explain better even as a short list.

Line 102: at the end of the sentence, I would suggest mentioning Table 1, which summarized all patients’ characteristics.

Paragraph 2.2.

Lines 104-108: why not add T lymphocyte baseline levels in Table 1, together with all the patients’ characteristics?

Lines 111-112: you state “measurements we defined as those taken within 4 weeks before receiving ICIs”. Is this specified elsewhere? Otherwise, please explain.

Lines 119-124: are there specific guidelines for irAEs (i.e. ASCO)? Please specify.

Results.

Paragraph 3.1.

I would suggest removing this paragraph or organizing/entitling differently.

In my opinion, patients’ characteristics should be described all in Paragraph 2.1, is there any particular reason why you put among results?

Lines 143-145: here you are mentioning adverse events, and you explain again irAEs in Paragraph 3.2, I think you could include them in the same paragraph, for more clarity.

Paragraph 3.2.

I would suggest writing this paragraph in a more structured way, as it is a bit confusing to me. Please describe briefly the various adverse events, as presented in Figure 1 and Table 2.

Lines 161-164: you mention “one type, multiple irAEs and grade ≥3”, I would suggest keeping the common classification, as grade 1, 2, ≥3.

Moreover, I think you should cite also grade 1 and 2 AEs, even briefly, in the text.

Paragraph 3.3.

Lines 196-198: is this the same as what you write in lines 185-187? Please verify.

Discussion.

Lines 315-317: have you assessed the presence of T cell infiltrates in the tissues?

Lines 343-344: please revise the sentence, it is not clear to me.

Conclusions.

Please revise the English language.

Line 372: what are you meaning by “pretreatment peripheral CD8+ T lymphocytes”?

Author Response

Response to Reviewer 1 Comments

Point 1: Lines 41-43: as it is written it seems contradictory, I would suggest highlighting that it is the combination of high CD8+ T and immune-related adverse events to represent a prognostic tool for the assessment of clinical outcomes after immunotherapy.

Response 1: Thank you for your valuable advice. The sentense has been revised to “Furthermore, patients who had higher baseline CD8+ T lymphocytes and experienced irAEs could have the longer PFS (18.4 months vs 2.2 months, P < 0.001) and OS (32.8 months vs 9.0 months, P = 0.001) than those who had lower CD8+ T lymphocytes and no irAEs.”

Point 2: Lines 45-47: please write differently, because it is a little confusing for readers.

Response 2: Thank you for your comment. The sentense has been revised to “Our study highlights the value of baseline peripheral CD8+ T lymphocytes as a predictive factor for irAEs in advanced NSCLC patients receiving ICIs. In addition, patients who have higher baseline CD8+ T lymphocytes at baseline and experience irAEs would have superior PFS and OS.”

Point 3: Line 66: what about grade 1 and 2 adverse events? Please write just a few words to mention.

Response 3: Thank you for your valuable comment. We have revised the sentence to “The incidence of all grade irAEs (according to Common Terminology Criteria for Adverse Events (CTCAE)) has been reported from 58% to 69%. Most of the irAEs are mild or moderate (grade 1 or grade 2), and the incidence of grade 3-5 irAEs is about 7%-13% in NSCLC.”

Point 4: Lines 83-86: please review the English language.

Response 4: Thank you for your advice. We have revised the sentence to ”Baseline peripheral lymphocytes have shown a great potential as a predictor of irAEs because of the advantages in practically noninvasive and dynamic monitoring. However, studies in this area are still limited.”

Point 5: Lines 83-86: is this coming from your observation or does it come from literature? In the latter, please cite the Reference.

Response 5: Thank you for your suggestion. Peripheral lymphocytes are blood biomarkers, so they have the advantage of noninvasive and dynamic detection.

Point 6: Line 99: which other therapies are you referring to? Please explain better even as a short list.

Response 6: Thank you for your valuable comment. We have revised the sentence to ”Treatment with anti-PD1/anti-PD-L1 drugs or anti-PD1/anti-PD-L1 drugs in combination with chemotherapy or antiangiogenesis therapy.”

Point 7: Line 102: at the end of the sentence, I would suggest mentioning Table 1, which summarized all patients’ characteristics.

Response 7: Thank you for your valuable comment. We have mentioned “Table 1” in this paragraph.

Point 8: Lines 104-108: why not add T lymphocyte baseline levels in Table 1, together with all the patients’ characteristics?

Response 8: Thank you for your valuable comment. Table 1 shows the categorical variables, and T lymphocyte baseline levels are continuous variables. We have added “The median baseline level of CD4+ T lymphocytes count was 266 (range 37-1008) M/L, the median baseline level of CD8+ T lymphocytes count was 288 (range 25-1195) M/L and the median baseline level of regulatory T lymphocytes count was 17 (range 3-63) M/L.” to describe T lymphocyte baseline levels in the manuscript.

Point 9: Lines 111-112: you state “measurements we defined as those taken within 4 weeks before receiving ICIs”. Is this specified elsewhere? Otherwise, please explain.

Response 9: Thank you for your valuable comment. Most clinical studies such as “Pacific 5” require that “the screening laboratory and imaging results must be obtained within 28 days prior to randomization”. So we also defined as baseline measurements within 4 weeks.

Point 10: Lines 119-124: are there specific guidelines for irAEs (i.e. ASCO)? Please specify.

Response 10: The NCCN guidelines were applied for management of immunotherapy-related toxicity.

Point 11: Paragraph 3.1. I would suggest removing this paragraph or organizing/entitling differently. In my opinion, patients’ characteristics should be described all in Paragraph 2.1, is there any particular reason why you put among results?

Response 11: Thank you for your valuable comment. I have moved this section to paragraph 2.1.

Point 12: Paragraph 3.2. Lines 143-145: here you are mentioning adverse events, and you explain again irAEs in Paragraph 3.2, I think you could include them in the same paragraph, for more clarity.

Response 12: Line 143-145 mentions different immune checkpoint inhibitors.

Point 13: Paragraph 3.2. I would suggest writing this paragraph in a more structured way, as it is a bit confusing to me. Please describe briefly the various adverse events, as presented in Figure 1 and Table 2.

Response 13: Thank you for your valuable comment. We have rewritten this paragraph.

“A total of 55 patients (50.5%) experienced at least one type of irAEs, and 38 patients (34.9%) developed multiple irAEs. Ninty-three patiens (85.3%) experienced mild and/or moderate (grade 1 and/or grade 2) irAEs, while grade 3-5 irAEs occurred in 16 patients (14.7%) in the study. Treatment related death was observed in 2 patients (1.8%). A total of 148 irAEs were observed in all patients, and 104 irAEs (70.3%) were resolved by the end of the observation. The most common any-grade irAEs were rash (16.5%), hypothyroidism (15.6%), pruritus (13.8%), pneumonitis (12.8%), hyperthyroisisn (11.9%), alanine transaminase increase (11.9%), gamma-glutamyltransferase increase (11.9%), aspartate aminotransferase increase (9.2%) and pyrexia (6.4%). The most common grade 3-5 irAEs were pneumonitis in 4 patients (3.7%) and gamma-glutamyltransferase increase in 3 patients (2.8%) (Fig. 1, Table 2). Median time to onset was 6.9 weeks and median time to resolution was 3.4 weeks for all irAEs (Fig. S1). The majority of new irAEs occurred from cycles 1 to 3 in all patients, and few new events occurred after 15 months (Fig. S2).”

Point 14: Lines 161-164: you mention “one type, multiple irAEs and grade ≥3”, I would suggest keeping the common classification, as grade 1, 2, ≥3.

Moreover, I think you should cite also grade 1 and 2 AEs, even briefly, in the text.

Response 14: Thank you for your valuable comment. We have revised the sentence to “A total of 55 patients (50.5%) experienced at least one type of irAEs, and 38 patients (34.9%) developed multiple irAEs. Ninty-three patiens (85.3%) experience mild and/or moderate (grade 1 and/or grade 2) irAEs, while grade 3-5 irAEs occurred in 16 patients (14.7%) in the study.”

Point 15: Lines 196-198: is this the same as what you write in lines 185-187? Please verify.

Response 15: Thank you for your valuable comment. As different statistical methods were used in Line 196 and Line 185, the results were different.

Point 16: Lines 315-317: have you assessed the presence of T cell infiltrates in the tissues?

Response 16: We have not evaluated the presence of T cell infiltrates in the tissues, and Lines 315-317 is a discussion of the literature 32-34.

Point 17: Lines 343-344: please revise the sentence, it is not clear to me.

Response 17: Thank you for your valuable advise. We have revised the sentence to “Furthermore, existing evidences have showed the prognosis value of irAEs in NSCLC patients treated with ICIs.”

Point 18: Line 372: what are you meaning by “pretreatment peripheral CD8+ T lymphocytes”?

Response 18: Thank you for your valuable advise. We have revised “pretreatment” to “baseline”.

Reviewer 2 Report

This study found that the baseline level of peripheral CD8+ T lymphocytes was the independent risk factor of the onset of irAEs, and was associated with longer survival in advanced NSCLC patients treated with PD-1/PD-L1 inhibitors. The combinational predictive value of baseline CD8+ T lymphocytes and the onset of irAEs for the clinical outcomes was also demonstrated. It’s a well-written study and only a minor issue is listed.

Comment 1

I am wondering the association of irAEs and peripheral CD8+ T lymphocytes in more detail. For example, is there a linear relationship between the occurrence rate or types of irAEs and the absolute value of peripheral CD8+ T lymphocytes? Please explore it as you can.

Reviewer 3 Report

The authors presented that baseline CD8+ T lymphocytes could be use as predictive marker for irAEs in NSCLC patients receiving ICIs.

Comments:

11.       In the Introduction section, the authors cite too many studies at once without explanation in more details. For example

“In recent years, immune checkpoint inhibitors (ICIs) including antibodies to programmed cell death-1 (PD-1) and programmed cell death ligand 1 (PD-L1) have revolutionized the treatment of several types of malignancy [3-6], including advanced NSCLC [7,8]”

It could be change to “… several types of malignancy including head and neck [3], Hepatocellular carcinoma [4], Oesophageal carcinoma [5] and so on

 2.       The authors need to explain in detail about grade of irAEs.

According to the introduction part, the authors mention that

“The incidence of all grade irAEs has been reported between 58% and 69%, and 65 the incidence of grade 3 or higher irAEs was found to be 7%-13% in NSCLC “

The question is “what is grade 3?, is it severe condition of irAEs? How’s about another grade”

In result part, Figure 1 shows 5 grade of irAEs.

So, the authors need to explain, it is important to readers for more understanding your point.

33.       There is much content overlapping between introduction and discussion. For example,

In Introduction, authors wrote the sentence, “higher irAEs was found to be 7%-13% in NSCLC” In Discussion, authors presented the information of “grade 3 or higher irAEs may still be observed in 7-13% patients”. They should recheck and modify Introduction and Discussion.

44.       Page 4, Line 150-151,

In the sentence “most had no or undetected common driver gene mutation (91.7%)”

Because there are several other driver gene mutations in NSCLC such as KRAS, BRAF, Met, etc. If only EGFR/ALK/ROS had been detected in this study. The authors should mention the name of driver gene mutations.

55.       I wonder why informed consent can be obtained from all patients because this is retrospective study using recorded data from 2017 to 2020.

Author Response

Response to Reviewer 3 Comments

Point 1: In the Introduction section, the authors cite too many studies at once without explanation in more details. For example

“In recent years, immune checkpoint inhibitors (ICIs) including antibodies to programmed cell death-1 (PD-1) and programmed cell death ligand 1 (PD-L1) have revolutionized the treatment of several types of malignancy [3-6], including advanced NSCLC [7,8]”

It could be change to “… several types of malignancy including head and neck [3], Hepatocellular carcinoma [4], Oesophageal carcinoma [5]” and so on

Response 1: Thank you for your valuable comments. We have revised the sentence to “In recent years, immune checkpoint inhibitors (ICIs) including antibodies to programmed cell death-1 (PD-1) and programmed cell death ligand 1 (PD-L1) have revolutionized the treatment of several types of malignancy including head and neck carcinoma [3], hepatocellular carcinoma [4], oesophageal carcinoma [5], colorectal carcinoma [6] and NSCLC [7,8].”

Point 2:  The authors need to explain in detail about grade of irAEs.

According to the introduction part, the authors mention that “The incidence of all grade irAEs has been reported between 58% and 69%, and the incidence of grade 3 or higher irAEs was found to be 7%-13% in NSCLC “

The question is “what is grade 3?, is it severe condition of irAEs? How’s about another grade”

In result part, Figure 1 shows 5 grade of irAEs.

So, the authors need to explain, it is important to readers for more understanding your point.

Response 2: Thank you for your valuable advise. We have revised the sentense. “The incidence of all grade irAEs (according to Common Terminology Criteria for Adverse Events (CTCAE)) has been reported from 58% to 69%, and the incidence of grade 3-5 irAEs was found to be 7%-13% in NSCLC [12-15].”

Point 3: There is much content overlapping between introduction and discussion. For example,

In Introduction, authors wrote the sentence, “higher irAEs was found to be 7%-13% in NSCLC” In Discussion, authors presented the information of “grade 3 or higher irAEs may still be observed in 7-13% patients”. They should recheck and modify Introduction and Discussion.

Response 3: Thank you for your valuable advise. We have rechecked and modified Introduction and Discussion.

Point 4: Page 4, Line 150-151,

In the sentence “most had no or undetected common driver gene mutation (91.7%)”

Because there are several other driver gene mutations in NSCLC such as KRAS, BRAF, Met, etc. If only c had been detected in this study. The authors should mention the name of driver gene mutations.

Response 4: Thank you for your suggestion. We have revised the sentense.

“Most patients had adenocarcinoma (55.0%), and most had no or undetected common driver gene (including epidermal growth factor receptor (EGFR), anaplastic lymphoma kinase (ALK) and V-ros UR2 sarcoma virus oncogene homolog 1 (ROS1)) alterlation (91.7%).”

Point 5:  I wonder why informed consent can be obtained from all patients because this is retrospective study using recorded data from 2017 to 2020.

Response 5: Thank you for your valuable advice. Because some of the patients have died or lost to follow up, we obtained informed consent from all patients who could be followed up. We have revised the sentence to “Informed consent was obtained from all patients who could be followed up in the study.”.

Round 2

Reviewer 3 Report

The manuscript has been sufficiently improved and is now ready to be published.